# Effect of Composite Edible Coatings Combined with Modified Atmosphere Packaging on the Storage Quality and Microbiological Properties of Fresh-Cut Pineapple

**DOI:** 10.3390/foods12061344

**Published:** 2023-03-22

**Authors:** Xingmei Liao, Yage Xing, Xiangfeng Fan, Ye Qiu, Qinglian Xu, Xiaocui Liu

**Affiliations:** 1Food Microbiology Key Laboratory of Sichuan Province, Chongqing Key Laboratory of Speciality Food Co-Built by Sichuan and Chongqing, College of Food and Bioengineering, Xihua University, Chengdu 610039, China; 2Key Laboratory of Food Non Thermal Processing, Engineering Technology Research Center of Food Non Thermal Processing, Yibin Xihua University Research Institute, Yibin 644004, China

**Keywords:** fresh-cut, pineapples, modified atmosphere packaging, edible coating, shelf life

## Abstract

This study investigated the effect of edible coating (EC), modified atmosphere packaging (MAP), and edible coating + modified atmosphere packaging (EC + MAP) treatments on the quality of fresh-cut pineapples during storage at 4 °C. The quality differences were analyzed by measuring the quality, physiological indicators, and total microbial counts. After 8 d of storage, the brightness (*L**) values of the EC + MAP and control samples were 72.76 and 60.83, respectively. The water loss and respiratory rate of the EC + MAP were significantly inhibited from 0% and 29.33 mg CO_2_ kg^−1^ h^−1^ to 4.13% and 43.84 mg CO_2_ kg^−1^ h^−1^, respectively. Furthermore, the fresh-cut pineapples treated with EC + MAP presented a good appearance, with lower total soluble solids (TSS) and relative conductivity and higher titratable acid (TA), ascorbic acid (AA), total phenol content, and firmness compared to the other treatment groups. At the end of storage, the EC + MAP samples exhibited the lowest polyphenol oxidase (PPO) activity, peroxidase (POD) activity, and malondialdehyde (MDA) content at 28.53 U, 60.37 U, and 1.47 nmol·g^−1^, respectively. Furthermore, the efficiency of EC + MAP treatment exceeded that of EC or MAP alone, preventing key problems involving the surface browning and microbiological safety of the fresh-cut pineapples. The results showed that EC + MAP treatment was more successful in maintaining the storage quality and extending the shelf life of fresh-cut pineapples.

## 1. Introduction

Pineapples (*Ananas comosus* L. Merr) are perennial plants belonging to the bromeliad family and are the third most traded tropical fruits in the world after bananas and mangoes [1]. They are popular with consumers due to their crown buds on top of the fruit, golden flesh, distinctive aroma, and sweet and sour taste. Pineapples are rich in nutrients, consisting mainly of carbohydrates and water, and are important sources of dietary fiber, sugar, organic acids, vitamins, and minerals [2]. The thick skin of the pineapple makes it difficult to eat and the large storage space leads to high transport costs. In recent years, the domestic and international markets for fresh-cut fruits and vegetables have expanded dramatically due to increasing consumer demand for fresh, convenient, additive-free, minimally processed, and nutritionally safe produce [3]. Fresh-cut pineapples are popular with consumers due to their nutritional value, freshness, convenience, and unique flavor. However, peeling and cutting can lead to mechanical damage, resulting in tissue juice exudation, which promotes the browning of the cut surface, nutrient loss from softened tissues, and spoilage, significantly reducing the nutritional and commercial value of the fruit [4]. Therefore, preserving and extending the shelf life of fresh-cut pineapple has attracted considerable research attention both domestically and abroad.

Edible coating (EC) preservation technology is considered promising for improving the quality of fruit and vegetables since it is safe, environmentally friendly, cost-effective, and simple to use [5]. Related studies have shown that composite coating films are more effective than single-coating films in preserving freshness, attracting significant research interest [6]. Sodium alginate is a natural polysaccharide able to concentrate solutions and form gels and films to extend the shelf life of fresh-cut fruit and vegetables. However, single sodium alginate coating films present the disadvantage of poor antibacterial and antioxidant properties [7]. Therefore, it can be combined with other materials to form a protective barrier due to its unique colloidal nature. Citric acid (CA) is an organic acid that can act as a chelating agent with isoascorbic acid and its neutral salts, providing an acidic environment that facilitates sterilization and preservation, anti-browning, and quality stabilization, consequently extending the shelf life of fresh-cut fruits and vegetables [8,9]. Sodium isoascorbate is a safe and effective antioxidant preservative with a strong reduction effect, which is widely used to inhibit browning in fresh-cut produce such as aubergines and peaches [10]. Therefore, combining sodium alginate, CA, and sodium isoascorbate creates a synergistic effect to improving fruit quality during storage.

The packaging process is a key step in producing, processing, and preserving fresh-cut pineapple products. Since consumers have become more health-conscious, residue-free, green preservation techniques are increasing in popularity. Since the 1970s, modified atmosphere packaging (MAP) technology has been used commercially to extend the shelf life of minimally processed fresh fruits and vegetables [11]. This technology has met the needs of people in pursuit of a healthy, green, natural, nutritious, and fast-paced lifestyle, while its efficacy and safety in preserving fruits and vegetables are widely recognized. MAP preservation has been extensively applied to fresh-cut apples, pears, carrots, papayas, and melons to extend the storage time of fresh produce and its quality [12]. MAP extended the shelf life of fruits and vegetables by changing the CO_2_ and O_2_ gas component ratio in the storage environment to inhibit respiration, microbial growth, and physiological metabolism and slow the aging process [13]. MAP has been widely used for preservation since it maintains product freshness, reduces storage losses, is simple to apply, and prevents environmental pollution. Siddiq et al. [14] reported that the color values of fresh-cut pears were affected after combined treatment with MAP and 2% NatureSeal at 4 °C, preserving the organoleptic quality of the fruit for 10 d. In addition, Waghmare et al. [15] investigated the combined effect of chitosan-coated film and aerated packaging treatment on the quality of fresh-cut papaya, revealing a shelf life extension of up to 25 d. These studies have shown that MAP effectively preserves fresh-cut fruit and vegetables. However, single MAP treatments are limited in maintaining the quality and shelf life of fresh-cut produce, and better results can be achieved when combined with other preservation techniques.

Several studies combined EC and MAP techniques to investigate the synergistic effect on fruit characteristics. Most existing studies on fresh-cut pineapple preservation used a single preservation technique for a single problem. Minimal research is available regarding the preservation of fresh-cut pineapples by combining MAP with composite EC films. Therefore, this study examines the effect of different treatments on the storage quality of fresh-cut pineapples by combining composite EC with MAP and analyzing the water loss, firmness, appearance, brightness (*L**) value, total soluble solids (TSS), titratable acid (TA), ascorbic acid (AA), total phenols, respiration rate, relative conductivity, malondialdehyde (MDA), polyphenol oxidase (PPO), peroxidase (POD), and the total number of bacteria colonies. This study aims to establish an integrated method to extend the shelf life of pineapples and promote the development of the fresh-cut pineapple industry.

## 2. Materials and Methods

### 2.1. Materials

Pineapples (*Tainong No.16*), with full eyes and 70% to 80% yellow skin for yellow ripening and free of diseases and mechanical damage, were selected for the experiment. The fruits were purchased from local pineapple growers and transported to the postharvest laboratory at the University of Xihua. The sodium alginate was purchased from Lianyungang Tiantian Seaweed Industry Co. Ltd. (Lianyungang, China), the CA from Shandong Weifang Ensign Industry Co. Ltd. (Weifang, China), and the sodium D-isoascorbic acid from Shandong Zhucheng Huayuan Biotechnology Co. Ltd. (Zhucheng, China). All three are food-grade food additives and all reagents used in the experiments were analytically pure.

### 2.2. Treatment and Storage

The head of the pineapple was removed from the tail of the pineapple using a Zhizhong No.1 pineapple cutter made in Yangjiang, Guangdong, China. The outer skin was peeled and cut into four equal pieces along the axis, which were then sliced into 1 cm thick, evenly sized fan-shaped pineapple slices.

The samples were treated as follows: (1) Control group (CK): The fresh-cut pineapples were packed directly in plastic boxes and not exposed to treatment or a modified atmosphere. (2) EC group: The optimal formulation of EC was obtained through preliminary single-factor experiments and response surface tests [16]. The pineapples were soaked in a compound EC solution consisting of 0.78% sodium alginate + 0.16% CA + 1.03% sodium isoascorbate by a mass fraction for 2 min and packed without exposure to a modified atmosphere. (3) MAP group: The fresh-cut pineapples were sealed and packed while exposed to a modified atmosphere (4% O_2_ + 5% CO_2_ + 91% N_2_) [17]. (4) EC + MAP group: The pineapples were soaked in a complex EC solution consisting of 0.78% sodium alginate + 0.16% CA + 1.03% sodium isoascorbate by mass fraction for 2 min, after which they were drained and packed while exposed to a modified atmosphere (4% O_2_ + 5% CO_2_ + 91% N_2_). The pineapples were labeled and refrigerated at 4 °C. Samples were taken every 2 d to determine their quality and physiological parameters.

### 2.3. Determination of the Water Loss, Firmness, Appearance, and L* Values

The water loss of the pineapples before and after storage was recorded at 2 d intervals to assess the values in each treatment group. The results were expressed as the water loss percentage, which was calculated using Formula (1):(1)$$Water\;loss\;rate\;(\%)=\frac{A-B}{A}\times 100$$
where *A* is the mass of the fresh-cut pineapples before storage, and *B* is the mass after storage.

The firmness was determined using a TA-XT PLUS texture analyzer (Beijing Weixun Super Technology Instrument Technology Co., Beijing, China)., according to a method described by Xing et al. [18] with slight modifications. The pineapple slices were placed directly below the cylindrical probe (5 mm diameter, P/5) for puncture testing. The pre-, mid-, and post-measurement velocities were set to 5 mm/s, 2 mm/s, and 2 mm/s, respectively, while the trigger force was set to 5 g. Five points were selected randomly on the surfaces of the fresh-cut pineapples, and the flesh hardness was calculated as the average force (N).

A photo was taken of the labeled, packaged, fresh-cut pineapples using a mobile phone in a well-lit area to record their appearance.

The *L** values of the fresh-cut pineapple surfaces were measured using a colorimeter model WF32 (Shenzhen Weifu Photoelectric Technology Co., Ltd., Shenzhen, China). Before the measurement, the equipment was standardized by calibrating a standard black and white plate. Five different positions were selected on each pineapple to obtain a uniform color measurement, after which the *L** value was recorded.

### 2.4. Determination of the TSS, TA, and AA

The juice of the pineapple pulp in each group was squeezed through three layers of clean gauze and collected in a beaker for use. The TSS values were obtained and recorded using an A610 digital refractometer (Haineng (Jinan) Instrument Co., Jinan, China).

The TA was determined via the sodium hydroxide titration method, as described by Liu et al. [19], with slight modifications. Here, 10 g of the fresh-cut pineapple pulp was weighed, homogenized, fixed to 100 mL, and filtered after 30 min. Then, 20 mL of the filtrate was aspirated, followed by the addition of phenolphthalein. The amount of NaOH was recorded by dropping the sample with a calibrated NaOH solution until it was pink and did not fade for 30 s. The results were calculated as CA (conversion factor 0.064), while the TA was expressed in grams of acid per 100 g.

The AA content was determined using the 2,6-chloroindophenol titration method described by Tao et al. [20], with slight modifications. The fresh-cut pineapples were crushed, homogenized, and filtered through three layers of gauze to obtain a filtrate. Next, 20 mL of the filtrate was placed in a conical flask and titrated with 2,6-dichloroindophenol until pale red, and the color did not fade for 30 s. Results were expressed as mg/100 g of sample.

### 2.5. Determination of the Respiratory Intensity and Relative Conductivity

An RS-CO2-N01-C portable headspace analyzer (Weihai Jingxun Changtong Electronic Technology Co. Weihai, China) was used to determine the respiratory intensity of the fresh-cut pineapples by making minor modifications to the method delineated by Liu et al. [21]. The sample mass (*m*_0_ g) was recorded, and the volume fraction of the CO_2_ in the sealed box (*A*_1_/%) was measured. The samples remained sealed for 1 h, after which the volume fraction of the CO_2_ in the sealed box was measured again (*A*_2_/%). The respiratory intensity was expressed as the mass of the CO_2_ released per kg of fruit and vegetables per h and calculated using Formula (2):(2)$$Respiratory\;intensity\;(\mathrm{mg}/(\mathrm{kg}\times h))=\frac{(A_{2}-A_{1})\times V\times M\times 1000}{V_{0}\times m_{0}\times t}\times 100$$
where *V* is the total volume in the sealed box in mL, *M* is the molar mass of the CO_2_ at 44 g/mol, *V*_0_ is the molar volume of the CO_2_ at 22.4 L/mol, and *t* is the measurement time in h.

The relative conductivity was determined according to a method described by Li et al. [22] with minor modifications. Here, 20 slices of the same size were collected from a fresh-cut pineapple sample using a hole punch, added to 40 mL of distilled water, and left to soak for 30 min. The solution was shaken with a vortex shaker for 20 min, after which the conductivity of the tissue solution (*L*_1_) was measured using a conductivity meter. The solution was then heated at 100 °C for 15 min, removed, and cooled to room temperature, after which the conductivity (*L*_2_) was measured. The relative conductivity (*L*) was calculated using Formula (3):(3)$$L=\frac{L_{1}}{L_{2}}$$

### 2.6. Determination of the Total Phenol and MDA Content

The total phenol content was determined using the colorimetric forintanol method described by Makroo et al. [23] with slight modifications. The standard curve was prepared using gallic acid. Here, 5.00 g of the fresh-cut pineapple sample was accurately weighed and placed in a 50 mL centrifuge tube, after which 30 mL of 70% ethanol solution was added. The mixture was centrifuged at 10,000 r/min for 5 min at 4 °C, and the supernatant solution was collected. Next, 1 mL of the ten-fold diluted sample solution was placed in a 25 mL test tube, after which 0.5 mL of Forinol reagent and 6 mL of distilled water were added. Finally, 4.00 mL of a 10.6% sodium carbonate solution was added, shaken well, left to stand for 2 h, and diluted to scale with distilled water. The absorbance was measured at 760 nm using distilled water as a control.

The *MDA* content was determined using the thiobarbituric acid (TBA) method, as delineated by Fan et al. [24], with slight modifications. Here, 1.0 g of the pineapple sample was mixed with 5 mL of a 100 g/L TCA solution, after which the homogenate was ground and centrifuged at 4 °C 10,000 r/min for 20 min. Next, 2.0 mL of supernatant (2.0 mL of a 100 g/L TCA solution was added to the blank tube) was mixed well with 2.0 mL of a 0.67% TBA solution. The mixture was boiled in a water bath for 20 min, cooled, and centrifuged again for 20 min. The supernatant was collected, and the absorbance values were obtained at 450 nm, 532 nm, and 600 nm. The calculations were performed using Equation (4):(4)$$MDA\;content(\mathrm{nmol}/g)=\frac{[6.452\times (OD_{532}-OD_{600})-0.559\times OD_{450}]\times V_{t}}{M\times V_{s}}$$
where *V_t_* represents the volume of the extract solution (mL), *V_s_* denotes the volume of the extract solution in the reaction mixture (mL), and *M* denotes the mass of the sample (g).

### 2.7. Determination of the PPO and POD

The PPO and POD activities were determined spectrophotometrically using a slightly modified method described by Wei et al. [25].

Enzyme extraction solution: in an ice bath, 5.0 g of the sample tissue was homogenized in 5.0 mL of extraction buffer (containing 1 mmol PEG, 4% PVPP, and 1% Triton X-100). The solution was centrifuged at 4 °C for 30 min at 100,000 r/min, and the supernatant was collected as the enzyme extract.

The PPO activity was determined at 420 nm using a UV240 UV/VIS spectrophotometer (Shanghai Sunyu Hengping Scientific Instruments Co., Shanghai, China). Here, 4.0 mL of 50 mmol/L acetic acid-sodium acetate buffer and 1.0 mL of a 50 mmol/L catechol solution were added to a test tube. This was followed by adding 100 puL of enzyme extract, and timing was started immediately. The POD activity was determined at 470 nm using a UV240 UV/VIS spectrophotometer. Next, 3.0 mL of a 25 mmol/L guaiacol solution and 0.5 mL of the enzyme extract were added to 200 uL of a 0.5 mol/L H_2_O_2_ solution to start the reaction. The absorbance change was measured at 420 nm and 470 nm every 30 s for 3 min at room temperature for PPO and POD, respectively. A unit of enzyme activity was defined as the amount of enzyme required to facilitate an absorbance change of 0.001 per min.

### 2.8. Microbiological Analysis

The total microbial count of the fresh-cut pineapples was evaluated according to a method described by Yousuf and Srivastava [26] with slight modifications. Here, 10 g of treated fresh-cut pineapples were added to 90 mL of sterile physiological saline, homogenized, and prepared as a 1:10 (g/mL) dilution at a sequential gradient. The whole procedure was performed on an ultra-clean table. Next, 1 mL of the solution at different dilution levels was collected in a sterilized plate, after which sterilized agar medium was added, shaken well, and left to stand. After solidification, the mixture was incubated upside down in a constant humidity incubator at 37 °C for 48 h. The results were expressed as logarithmic values of the total number of colonies.

### 2.9. Statistical Analysis

The experimental results were counted using Excel 2019 software (Microsoft Corporation, New York, NY, USA) and plotted using Origin 2019 software (Stat-Ease Inc., Minneapolis, MN, USA). The data were subjected to Duncan’s ANOVA (*p* < 0.05 indicated a significant difference) using SPSS 20 software (IBM Corporation, New York, United States). The analyses were repeated three times, and the results were expressed as mean ± standard deviation.

## 3. Results and Discussion

### 3.1. Water Loss Rate, Firmness, Appearance, and L* Values

The water loss rate of the fresh-cut pineapples after different treatments during storage is shown in Figure 1a. The water loss rate of all the samples increased continuously with extended storage time. From 4 d onward, the water loss rate was significantly (*p* < 0.05) lower in the EC, MAP, and EC + MAP treatment groups than in CK. The EC and EC + MAP groups displayed lower water loss rates than the uncoated CK and MAP groups throughout the storage period. At the end of storage, the water loss of both the coated groups did not exceed 5%, indicating that coating the surfaces of the fresh-cut pineapples effectively restricted water loss. The EC + MAP group displayed the lowest water loss, followed by the EC and MAP groups, while CK exhibited the highest water loss of 6.87%. As shown in Figure 1b, the hardness values of all the samples gradually decreased during storage. EC, MAP, and EC + MAP displayed significantly (*p* < 0.05) higher hardness values than CK from 2 d onward. After 4 d, the EC + MAP group presented the highest hardness values (*p* < 0.05), followed by EC and MAP.

These results indicated that the combined EC + MAP treatment was more effective in maintaining the weight and appropriate firmness of the fresh-cut pineapples during storage than the individual treatments. The CK group showed the most significant water loss decline during storage since cutting the fresh pineapples increased the area in contact with air, leading to more severe dehydration. The semi-permeable membrane formed by the sodium alginate coating on the surface of the fresh-cut pineapple acted as a physical barrier against gas and water exchange, minimizing moisture and water loss during storage [27]. Azarakhsh et al. [28] reported that sodium alginate ECs maintained the firmness of fresh-cut pineapples by significantly (*p* < 0.05) restricting the water loss decline. Since MAP technology retains water via packaging, it reduces water loss during transpiration [29]. Similar results were obtained by Oguz-Korkut et al. [30] for fresh-cut ‘*Deveci*’ pears. MAP packaging retained a high relative humidity at 4 °C and 4% O_2_ + 5% CO_2_, maintaining the firmness and water of the sliced pears.

The appearance of the fresh-cut pineapple samples after the four different treatments during storage at 4 °C is shown in Figure 1c. EC + MAP treatment was the most effective in increasing the appearance and color of the samples, while the CK group displayed severe browning. The *L** value, denoting the color changes in the fresh-cut pineapples, decreased in all the samples throughout the storage period (Figure 1d). From 2 d to 8 d, the *L** value of the CK group decreased significantly (*p* < 0.05), from 72.16 to 60.83. Compared to the treated samples, the untreated samples showed a more substantial *L** value decrease of up to 17.37%, while the MAP, EC, and EC + MAP groups displayed a less dramatic decline of 10.03%, 9.02%, and 4.41%, respectively. Therefore, the EC, MAP, and EC + MAP treatments were all effective in protecting the color of the fruit samples.

The color changes in the pineapple samples were mainly related to oxidation processes in response to the energy required to metabolize the products. EC + MAP treatment maintained the *L** values of the pineapple samples at a consistently high level (70.12–74.53) during storage, indicating that the combined treatment was more successful in maintaining the original color. Although refrigeration temperatures slowed fruit browning, it became more pronounced as the respiration rate and storage time increased (without MAP). However, studies have shown that single MAP treatment is not sufficient in preventing enzymatic browning and requires anti-browning treatments [31]. In this study, combining EC with MAP effectively prevented tissue browning and spoilage for up to 16 d. However, it is worth pointing out that in order to evaluate the browning of fresh-cut pineapples more comprehensively and specifically, we should measure the difference in the color difference ΔE, which should be improved in subsequent studies. The color change of fresh-cut pineapple was related to the semi-permeability of the cell membrane. When the cell membrane was damaged, pigmented material flowed into the intercellular space along with the intracellular fluid, changing the refractive index between the cell and its surroundings [32]. Siddiq et al. [14] found that the combined treatment of fresh-cut pears with MAP and calcium ascorbate effectively prevented tissue browning and extended their shelf life up to 21 d. EC and EC + MAP effectively reduced the *L** value changes in the pineapple samples. This could be ascribed to the protective effect of the composite coating on the cell membranes of pineapple tissue, reducing the flow of intracellular fluid and pigmented material.

### 3.2. The TSS, TA, and AA Content

The TSS content of fresh-cut pineapples after different treatments during storage is shown in Figure 2a, showing an increasing trend in all the groups. This could be attributed to the continued conversion of starch and acid metabolism to sugar during storage after cutting [33]. The TSS of the untreated samples was significantly higher than in the treated samples throughout the storage period (*p* < 0.05). At 8 d of storage, the TSS levels of the CK, EC, MAP, and EC + MAP groups were 18.68, 16.94, 17.17, and 16.75, respectively, an increase of 15.52%, 3.48%, 5.92%, and 2.01%, respectively, compared to 0 d. Therefore, EC, MAP, and EC + MAP treatment delayed the rise in TSS content and slowed the ripening of the fresh-cut pineapples. The pineapple samples treated with EC + MAP displayed significantly (*p* < 0.05) lower TSS levels after 12 d than individual EC and MAP treatments. Furthermore, the TSS increase was more gradual in the EC + MAP group. This may be due to the reduced respiratory activity and impaired metabolic activity of the samples [15]. Therefore, the composite treatment was more effective in maintaining the TSS content of the freshly cut pineapples.

The TA content of fresh-cut pineapple after different treatments during storage is shown in Figure 2b. The change patterns of the TA content in the four groups showed similar decreasing trends as the storage time was extended. At 0 d of storage, the TA values were higher in the EC and EC + MAP groups than in the CK and MAP groups, which could be ascribed to the addition of isoascorbic and CA to the EC. The TA of all the samples decreased during the first 12 d of storage due to the conversion of organic acids to sugars and their use as substrates for respiratory metabolism [34]. The TA content was significantly (*p* < 0.05) higher in the EC, MAP, and EC + MAP groups than in the CK group and considerably (*p* < 0.05) higher in the EC + MAP group than in the EC and MAP groups. A TA level was preferred in the fresh-cut pineapple samples during storage since it was associated with a low pH, consequently preventing the early growth of microorganisms [35]. Similarly, Passafiume et al. [36] showed that using three different ECs for fresh-cut kiwifruit yielded a similar trend in the TA levels when the storage time was extended, exhibiting a decrease in the organic acid content as the fruit matured. The TA decreased while the SSC increased, indicating the ripening index of the fruit. This was consistent with the findings of Pleșoianu et al. [37], who combined pectin-coated film with a chemical dip of AA + CA to treat fresh-cut pears. The results revealed that this treatment slowed down the decline in hardness, TA, and organic acid during storage, effectively maintaining the fruit quality. Therefore, EC + MAP effectively maintained the TSS and TA content, delaying the ripening and aging process of fresh-cut pineapples.

The changes in the AA content of the pineapple samples during refrigeration at different treatments are shown in Figure 2c, decreasing with extended storage time. The most significant decline of 36.14% was evident in the CK group at 8 d of storage. During the early stages of storage (0–2 d), the AA levels remained relatively stable, with no significant (*p* > 0.05) differences between the samples of the respective treatment groups and CK. At the end of storage, the AA levels in the EC (14 d), MAP (12 d), and EC + MAP (16 d) groups were 31.16 mg/100 g, 29.38 mg/100 g, and 34.60 mg/100 g, respectively, exhibiting a decrease of 19.40%, 25.73%, and 11.30% compared to the beginning of storage (0 d). The pineapple samples in the EC + MAP group exhibited higher AA levels during storage, partly because the surface coating restricted gas exchange and inhibited AA oxidation [38]. However, MAP treatment significantly (*p* < 0.05) delayed the decline in the AA content in the pineapple samples by inhibiting the physiological metabolism and reducing the oxidation and loss of AA by decreasing the O_2_ and CO_2_ in the gaseous environment to a suitable ratio [39].

### 3.3. Respiratory Intensity and Relative Conductivity

The respiration rate of the fresh-cut pineapples after different treatments is shown in Figure 3a. The respiration rate of the CK group continued to increase during storage, while that of the EC, MAP, and EC + MAP groups exhibited an initial increase, followed by a decrease or stabilization. From 2 d onwards, the respiration rate of the treatment group was significantly (*p* < 0.05) lower than that of the CK group. At 8 d of storage, the CK group showed the highest respiratory intensity at 82.34 mg/(kg·h) compared to 51.28 mg/(kg·h), 63.33 mg/(kg·h), and 43.84 mg/(kg·h) in the EC, MAP, and EC + MAP groups, respectively. During storage, the respiration rates of fresh-cut pineapples were significantly (*p* < 0.05) lower in the EC + MAP group than in the other treatment groups. This indicated that the EC + MAP group had a stronger ability to inhibit higher respiration rates. Respiratory intensity is a vital physiological metabolic factor, reflecting the effect of storage on the fruit, and its variation can indicate changes in the physiological activity of the sample [40]. In general, a higher respiratory intensity indicates faster nutrient consumption. EC + MAP displayed the strongest ability to inhibit the respiration rate of fresh-cut pineapple. This is consistent with the findings of Zhang et al. [41], who showed that film-coating materials (sodium alginate and chitosan) allowed the accumulation of CO_2_ produced during fruit respiration while reducing the inward flow of O_2_ within the fruit tissues to limit respiration. Furthermore, combined MAP treatment maintained a low relative humidity in the storage environment, decreasing intracellular water mobility to inhibit the respiration rate.

Relative conductivity is an important indicator for characterizing the integrity of the cell membrane, with higher values indicating more severe damage [42]. As shown in Figure 3b, the relative conductivity of all the samples showed an upward trend with the extension of storage time. From 2 d onward, the relative conductivity of the CK group increased rapidly and was significantly (*p* < 0.05) higher than the treated group, indicating that its cell membrane permeability increased and was oxidatively damaged. After 2 d, the relative conductivity of the CK group increased rapidly and was significantly (*p* < 0.05) higher than in the EC, MAP, and EC + MAP groups, indicating increased permeability and oxidative damage to their cell membranes. The initial relative conductivity of the EC group (17.20%) was significantly (*p* < 0.05) higher than the MAP group (12.39%). However, after 4 d, the relative conductivity of the EC group gradually declined to below that of the MAP group, indicating that EC treatment was more successful in maintaining the cell membrane permeability of fresh-cut pineapple than MAP treatment. Contrarily, by 12 d of storage, the relative conductivity of the EC + MAP group was 31.10%, which was significantly (*p* < 0.05) lower than the EC and MAP groups, delaying the rise in relative conductivity. This suggests that the EC + MAP group can effectively reduce the damage to the cell membranes of fresh-cut pineapples and maintain the relative conductivity at a low level. Similarly, Meng et al. [43] found that treating fresh-cut peppers with pressurized argon effectively restricted the rise in relative conductivity and reduced electrolyte leakage. The high argon level during EC + MAP treatment played a corresponding synergistic role in maintaining the integrity of the cell membrane structures of the pineapple samples.

### 3.4. Total Phenols and MDA Content

Phenolics display strong antioxidant effects that can effectively remove reactive oxygen species and inhibit membrane lipid peroxidation. They also play an important role in the plant antioxidant process [44]. As shown in Figure 4a, the total phenolic content in the fresh-cut pineapple displayed an initial increase, followed by a decrease during refrigeration after exposure to different treatments. During storage, the total phenolic substances in the treatment groups were significantly (*p* < 0.05) higher than in the CK group. The initial total phenolic content of the CK group was 21.38 mg/100 g, reaching a maximum value of 27.64% at 8 d of storage. Contrarily, the initial total phenolic content of the EC + MAP group was 21.87 mg/100 g, reaching a maximum value of 53.77% at 10 d of storage. Overall, the EC + MAP group increased the total phenolic content relative to the CK group, effectively promoting the synthesis and accumulation of phenolic substances. The cutting and processing of pineapples increase the exposure of polyphenols to oxygen, continuously decreasing the phenolic content when catalyzed by POD and PPO [45]. EC + MAP was more successful in delaying the reduction in the total phenolic content of the fresh-cut pineapple by inhibiting the PPO and POD activity and reducing phenolic oxidation, consequently maintaining a high total phenolic level.

MDA content changes can be used to delineate the cell membrane lipid peroxidation level in fruit in senescence or stress conditions [46]. As shown in Figure 4b, the MDA content in the CK group samples continued to increase during refrigeration, while EC, MAP, and EC + MAP treatment inhibited MDA levels. During the first 6 d of storage, EC + MAP treatment effectively restricted MDA accumulation in the pineapple samples compared to the other three groups, resulting in a slow increase. After 8 d of storage, the MDA content of the EC + MAP group was 1.25 nmol/g, which was significantly (*p* < 0.05) lower than the CK group, indicating significant lipid peroxidation and MDA accumulation in the CK group samples. The results showed that EC + MAP treatment successfully inhibited MDA accumulation and reduced oxidative damage during storage. Similarly, Xing et al. [47] used a chitosan coating and MAP containing anti-browning agents to treat fresh-cut lotus roots. Samples treated with the chitosan coating + MAP showed the lowest MDA content and the slowest increase, more successfully inhibiting browning and extending the shelf life of the fresh-cut lotus roots. Shen et al. [48] demonstrated that synergistic treatment (composite coating combined with MAP) significantly restricted the rise in MDA, respiration rate, and enzyme activity (PPO and POD). The results showed that EC + MAP treatment was most effective in controlling MDA accumulation and reducing oxidative damage during storage.

### 3.5. PPO and POD Activity

The impact of different treatments on the PPO activity in the fresh-cut pineapples is shown in Figure 5a. The PPO activity in the EC, MAP, and EC + MAP groups displayed an initial increase, followed by a decline. Both the EC and EC + MAP treatments significantly (*p* < 0.05) slowed a rise in PPO compared to the CK group. From 0–6 d of storage, the PPO activity increased rapidly in all the groups, while all three treatment groups inhibited the PPO activity to varying degrees after 6 d, resulting in a plateauing or decreasing trend. At 16 d of storage, the PPO activity level in the EC + MAP treatment group was only 28.53 U/min·g. As illustrated in Figure 5b, the POD activity of the pineapples in all the groups initially increased, followed by a stabilization or a decline during storage. The POD activity in all treatment groups was significantly (*p* < 0.05) lower than in the CK group at the end of storage. No significant (*p* > 0.05) differences were evident between the POD activity changes in the three treatment groups (EC, MAP, and EC + MAP) during the first 2 d of storage. At 8 d of storage, the POD activity in the EC and EC + MAP groups was 78.63 U/min·g and 63.38 U/min·g, respectively, which was significantly (*p* < 0.05) lower than in the CK group (105.74 U/min·g). The POD activity in the pineapple samples in the EC + MAP group was 63.74 U/min·g at 14 d of storage, which was considerably (*p* < 0.05) lower than in the EC group (72.17 U/min·g), indicating that the combined treatment more successfully inhibited the POD activity and maintained the storage quality of fresh-cut pineapples compared to individual treatment.

The PPO and POD activity was more successfully inhibited in the EC + MAP-treated pineapple samples than in the other groups. On the one hand, this was attributed to the anti-browning agent incorporated in the sodium alginate coating. As a browning inhibitor, isoascorbic acid has attracted significant research attention. Many studies have utilized sodium isoascorbate to protect the color of fresh-cut apples and mangoes, demonstrating that it effectively inhibits browning caused by PPO, POD, and other key enzymes [49,50]. CA exhibited an inhibitory effect on the elevated activity of browning-related enzymes. On the other hand, the combined treatment effectively inhibited PPO and POD activity, which was possibly related to the changes in the gas composition. MAP reduced the oxygen content of the air in the fresh-cut pineapple storage environment and increased the carbon dioxide content, consequently controlling respiration rate, enzyme activity, and oxidation [51]. Sortino et al. [52] reported that coating + MAP treatment significantly (*p* < 0.05) reduced the PPO activity and flesh browning in fresh-cut persimmons, which was vital for regulating postharvest rot and maintaining fruit quality. In addition, Augusto et al. [53] investigated the effect of natural alginate-based coatings and MAP on the quality of fresh-cut apples, showing that samples coated with seaweed extract and packaged via MAP showed lower PPO and POD activity. A more significant change in the PPO activity increased the browning rate, exhibiting a positive correlation between the two. The elevated enzyme activity could be ascribed to the conversion of PPO from a bound to a free state during storage due to post-ripening, senescence, stress, and membrane disruption, stimulating its activity [54]. O_2_ represents a key factor for inducing PPO reactivity with phenolic compounds. The MAP group restricted the oxidation of the pineapple pulp by reducing the oxygen concentration in fresh-cut pineapple [55]. In addition, AA inhibited PPO, while EC treatment slowed the oxidative loss of AA in the pineapple samples. Therefore, the PPO activity was lower in the EC and EC + MAP groups than in the CK group. POD is an oxidoreductase enzyme prevalent in fruits that scavenges reactive oxygen species by catalyzing the breakdown of peroxides and H_2_O_2_ [56]. Overall, EC + MAP treatment reduced the respiratory intensity of the pineapple and inhibited the PPO and POD activity to a certain degree, which delayed senescence and preserved freshness.

### 3.6. Microbiological Analysis

The impact of different treatments on the total aerobic microbial counts in the fresh-cut pineapple is shown in Figure 6. The total number of colonies increased with storage time after all treatments. At 0–2 d of storage, the microbial growth was slow in all groups. At 2–4 d of storage, the CK group showed active microbial growth and a linear increase in the total number of colonies in the samples, while this increase occurred gradually in the three treatment groups and was significantly (*p* < 0.05) lower than in the CK group. At 10 d of storage, the total number of colonies in the EC, MAP, and EC + MAP groups reached 4.02, 4.21, and 3.94 lg (CFU/g), respectively, while the CK group reached 6.45 lg (CFU/g). Toxic substances may be produced when the total number of colonies exceeds 6 lg (CFU/g) [57]. However, the CK group exceeded this safety threshold at 10 d, while the MAP, EC, and EC + MAP groups reached this level at 14 d, 16 d, and 18 d, respectively. Therefore, all three treatments further inhibited microbial growth in the fresh-cut pineapple. EC + MAP exhibited the most successful inhibitory effect and the longest storage time, followed by the EC, MAP, and CK groups with storage times of 16 d, 14 d, 12 d, and 8 d, respectively.

Since the aerobic bacterial count reflects the freshness of the product and the hygienic conditions of the production facility, microbiological safety is crucial for fresh-cut fruit and vegetables [58]. The CK group displayed the fastest microbial growth since the fresh-cut surfaces of the pineapple samples were exposed to air. However, MAP inhibited the growth of microorganisms by reducing the oxygen content. Compared to the CK and MAP groups, the EC more effectively inhibited microbial growth. Treatment with a mixture of sodium alginate, sodium isoascorbate, and CA in appropriate amounts facilitated the formation of a protective film on the surfaces of the fresh-cut pineapples, creating a low-oxygen microaerophilic environment within the film to effectively reduce oxygen infiltration and protect the color of the fruit [59]. Microbial growth was more successfully restricted in the EC + MAP group than in the other groups, highlighting the EC and MAP combination as an effective synergistic strategy for microbial inhibition and promoting the microbiological safety of fresh-cut pineapples.

## 4. Conclusions

This paper analyzes the impact of three different treatments on the storage quality and microbiological properties of fresh-cut pineapples. During storage, the fruit quality is more successfully retained in the EC + MAP group, followed by the EC, MAP, and CK groups. Combining EC with MAP may be useful in maintaining the quality of fresh-cut pineapples for up to 16 d of storage at 4 °C. Neither EC nor MAP alone can effectively address the key issues of browning and microbiological safety. The combined treatment strategy significantly improves the microbial safety of fresh-cut pineapples and effectively solves the browning problem by inhibiting the PPO and POD activity in the samples. Furthermore, the use of EC + MAP presents an excellent approach for reducing water loss, TSS, AA, and TA, while delaying increased respiratory intensity, relative conductivity, and MDA levels to maintain the fruit firmness. In comparison, the pineapple quality indicators in the EC + MAP group are higher than in the other treatment groups. This study provides a comprehensive strategy for the effective browning inhibition and extended storage life of fresh-cut pineapples.

## Figures and Tables

**Figure 1 foods-12-01344-f001:**
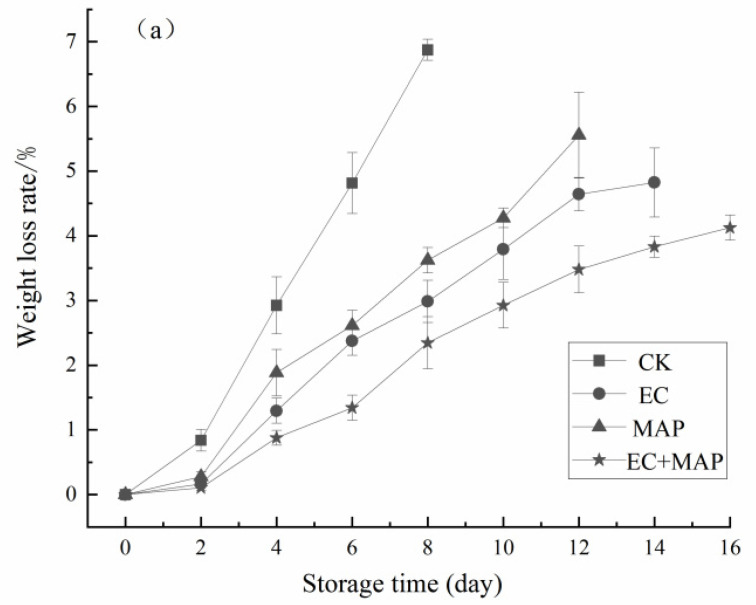
Effect of different treatments on water loss (**a**), firmness (**b**), appearance (**c**) and *L** values (**d**) of fresh-cut pineapples during storage.

**Figure 2 foods-12-01344-f002:**
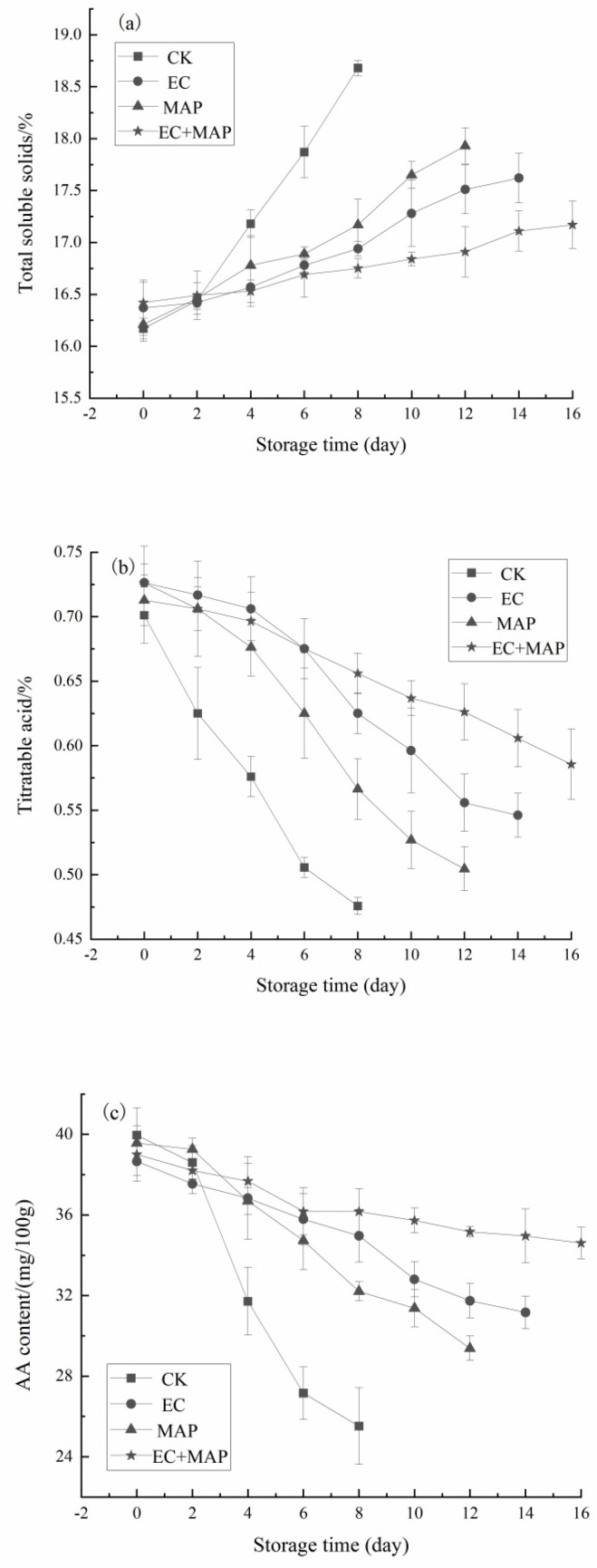
Effect of different treatments on TSS (**a**), TA (**b**) and AA (**c**) of fresh-cut pineapples during storage.

**Figure 3 foods-12-01344-f003:**
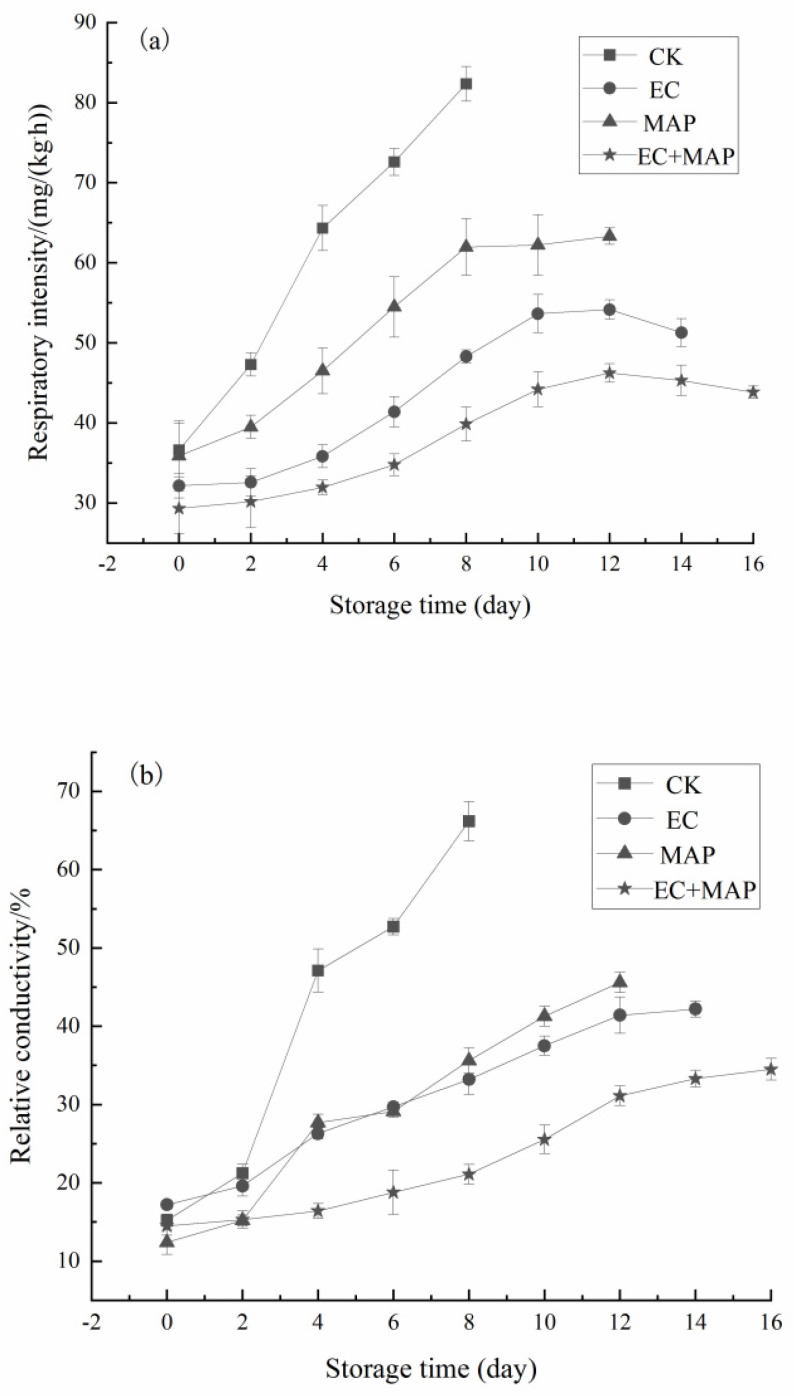
Effect of different treatments on respiratory intensity (**a**) and relative conductivity (**b**) of fresh-cut pineapples during storage.

**Figure 4 foods-12-01344-f004:**
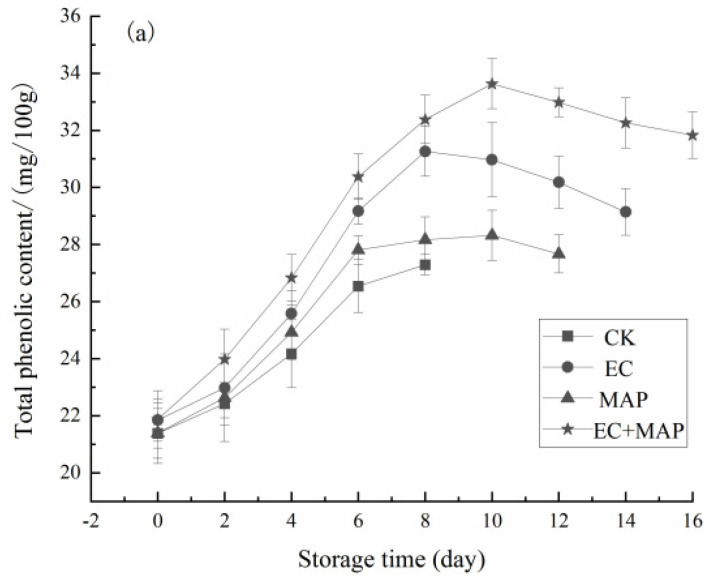
Effect of different treatments on total phenols (**a**) and MDA content (**b**) of fresh-cut pineapples during storage.

**Figure 5 foods-12-01344-f005:**
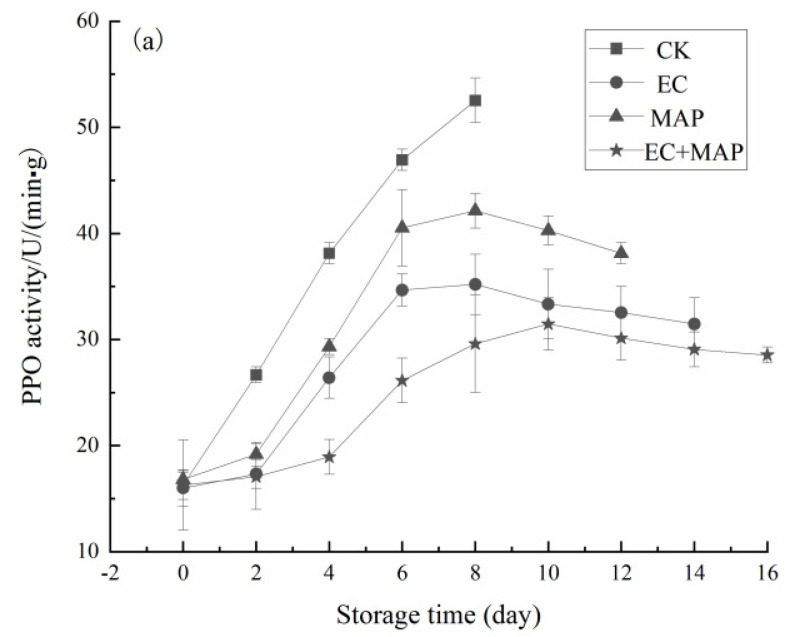
Effect of different treatments on PPO activity (**a**) and POD activity (**b**) of fresh-cut pineapples during storage.

**Figure 6 foods-12-01344-f006:**
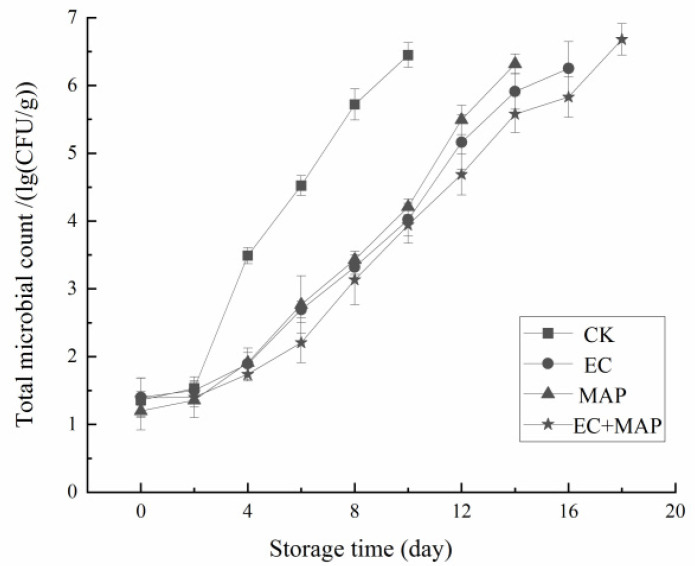
Effect of different treatments on total number of microorganisms of fresh-cut pineapples during storage.

## Data Availability

Not applicable.

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
