# Peer review of "Effect of Composite Edible Coatings Combined with Modified Atmosphere Packaging on the Storage Quality and Microbiological Properties of Fresh-Cut Pineapple"

_foods, 2023, doi:10.3390/foods12061344_

Round 1

Reviewer 1 Report

I am pleased to announce that the manuscript for review entitled "Effect of composite edible coatings in combination with modified atmosphere packaging on the storage quality and microbiological properties of freshly cut pineapple" is a very interesting study. The issues presented in the content are described clearly and with unquestionable knowledge of the subject. However, I believe that its diligence in formal terms should be improved - especially in terms of designations in formulas and their descriptions.

The text is written in understandable language, and the figures and information in the tables are clear and supported by scientific literature. Due to the multidimensionality of the discussed issues, it seems reasonable to present the issue more clearly, along with practical justification. There are studies whose practical conclusions are undeniable, also in these times of at least significant progress in the applied research techniques - please consider these studies, especially those whose conclusions are the basis for undertaking this type of scientific research, go so far and give them practical nature in summary.

However, I have a few comments:

Ad 2.5. Determination of the respiratory intensity and relative conductivity

line 171-177

1. please explain why conductivity was measured in solutions and not directly in pineapple fruit?

2. what can be concluded on the basis of such a measurement in terms of fruit storage in the prevailing test conditions?

Very interesting and valuable information related to the course of food processing (blanching, freezing, thawing and storage in changing conditions) can be provided by the analysis of electrical properties of food. These quantities are increasingly used in food technology. Their knowledge turns out to be helpful in assessing the authenticity and botanical origin of both plant and animal (honey) succulents (honey), it can be used to assess the fat content or the presence of somatic cells (milk) and provide information on the gluten content in flour. The increasing use of electrical properties is also related to the development of unconventional methods of processing agricultural and food products, e.g. resistance heating or pulsed electric field. Electrical parameters, especially conductivity electric power, determine the course of these processes. In addition, knowledge of the electrical conductivity is crucial when assessing the effectiveness of a structure on food.

Changes related to the damage to the cell membrane as a result of the pretreatments used may accelerate the maturation or leakage of intracellular content containing substances capable of accumulating an electric charge and, consequently, may lead to an increase in electrical conductivity. Therefore, it can be concluded that the measurement of the electrical properties of plant tissue can be a valuable tool for assessing the course of these processes and for indirect assessment of structural changes caused by such technological treatment. It is worth adding that the literature related to these issues is rather poor and, according to the authors' knowledge, is limited to a few items.

Ad 3.1. Weight loss, firmness, appearance, and L* values

line 238-239

I consider it wrong to use the term "...The weight loss rate...". The authors did not determine the rate of mass loss, but only determined the changes in mass over time. Changes in mass over time should be described by the kinetic equation and the first derivative should be determined, which would make it possible to determine the rate of mass change over time or as a function of the dry mass content. The speed should be related to the stability of the parameter, which is the content of dry substance mass, because changes in mass result from the loss of water from the test material. Determination of weight loss in % is misleading and does not allow to draw appropriate conclusions about the storage of pineapple pieces after initial protection under different conditions.

Please also explain (maybe in the methodology) why the fruit was stored for 16 days, how time and conditions were selected.

line 263-272

Please explain why the assessment of color changes was carried out solely on the basis of changes in brightness L of the tested samples. It seems reasonable to determine the absolute color difference DE, especially since the authors comment on color changes by mentioning browning, and the color coordinates +a (share of red color) and +b (share of yellow color) correspond to the change towards both enzymatic and non-enzymatic browning, which can be included in the calculations as saturation DC and/or hue angle H. The L parameter is only responsible for brightness.

I suggest determining the absolute color difference of DE compared to the color of fresh pineapple fruit.

In the L*a*b* color space there can be a color difference expressed as a single numerical value ΔE.

It should be remembered, however, that it only determines the size of the color difference, but does not inform what the difference is - this should be explained - and this is the task of the authors. ΔE is described by the following equation: ΔE = √(ΔL* )2+(Δa* )2+(Δb* )2.

Reviewer 2 Report

The study presents data on the impact of composite edible coatings in combination with modified atmosphere packaging for postharvest storage of fresh-cut pineapple. The topic aligns with the aims and scope of the journal and is engaging. The experimental design is robust, but it would benefit from further clarification on some points.

-L.93: change 'contribute' to 'aims'

-L.106: the pineapple was prepared using a specialized pineapple cutter. specify what type of specialized cutter was used.

-L.109: why this EC composite solution? More information must be provided.

-L.117: how was the freshly cut pineapple drained?

-All graphs presented in this work should have a mean separation.
